# Context-Aware Autoregressive Models for Multi-Conditional Image Generation

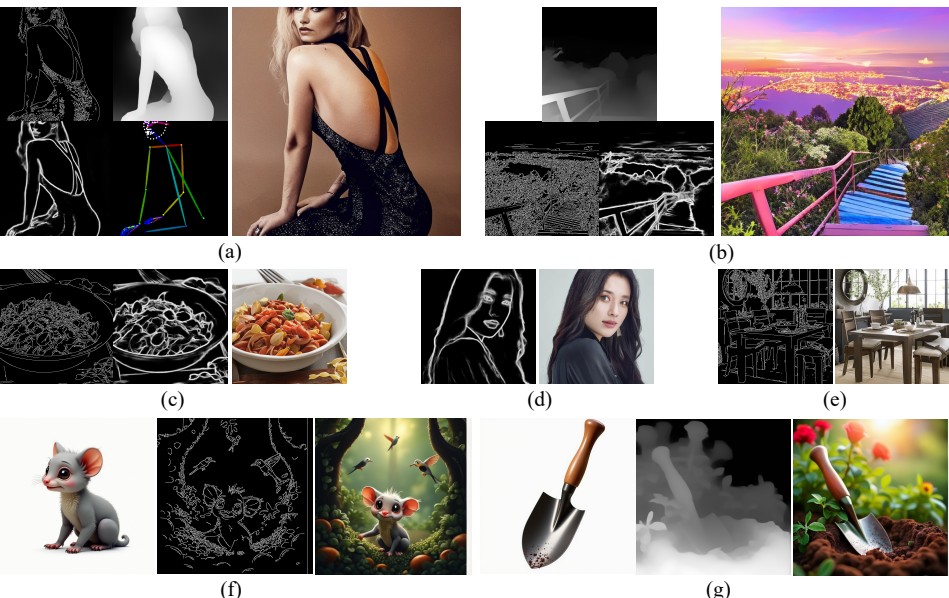

Figure 1: **Visualization of generated results from our proposed ContextAR framework.** The combinations of conditions are: (a) Canny + Depth + HED + Pose, (b) Canny + Depth + HED, (c) Canny + HED, (d) HED, (e) Canny, (f) Subject + Canny, (g) Subject + Depth. Our framework is implemented on an autoregressive model, achieving excellent controllability while offering remarkable flexibility and versatility.

## Abstract

Controlling generative models with multiple, simultaneous conditions is a critical yet challenging frontier. Mainstream diffusion models, despite their success in single-condition synthesis, often exhibit performance degradation and condition conflicts in this setting. We identify the root cause of this limitation as the inherent *parallel* generation process of these models. By applying all conditional constraints globally and concurrently, they create a "tug-of-war" between competing guidance signals, forcing suboptimal compromises. This paper advocates for a paradigm shift to *serial* generation. We posit that autoregressive models, by constructing images token-by-token, can resolve conflicting constraints locally and sequentially, enabling a more harmonious and precise integration of multiple conditions. To realize this paradigm, we introduce **ContextAR**, an autoregressive framework that represents diverse conditions within a unified sequence. It employs a novel Conditional Context-aware Attention mechanism that restricts inter-condition communication, enhancing both compositional flexibility and computational efficiency. Extensive experiments validate our hypothesis: ContextAR significantly outperforms state-of-the-art parallel (diffusion-based) methods in controllability and faithfulness to multiple conditions, without a trade-off in image quality. Our work establishes serial generation as a more powerful and flexible paradigm for the complex task of multi-conditional image synthesis.

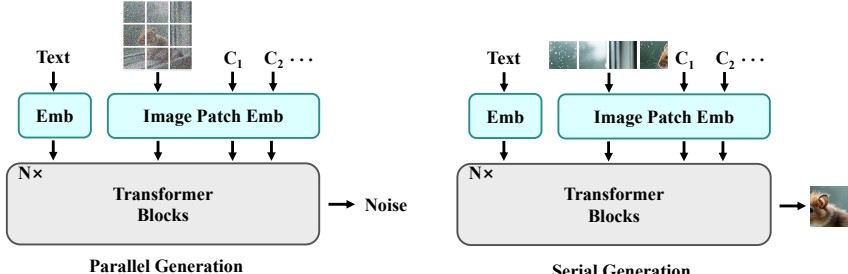

Figure 2: Comparison of parallel generation and serial generation in multi-condition tasks.

# 1 INTRODUCTION

The advent of diffusion models has marked a significant milestone in generative AI, enabling the synthesis of photorealistic and diverse images from simple text prompts (Ho et al., 2020; Song et al., 2020; Rombach et al., 2022). This breakthrough was soon followed by methods offering fine-grained, single-condition control. Frameworks like ControlNet (Zhang et al., 2023) and T2I-Adapter (Mou et al., 2024) empowered users to dictate specific spatial layouts, human poses, or object boundaries, providing a much higher degree of precision than text alone.

The natural next step for creative applications is to combine these controls, enabling users to specify multiple constraints simultaneously—for instance, defining a subject's appearance and the scene's layout at once. A growing body of research (Zhao et al., 2023; Hu et al., 2023; Wang et al., 2025a) has explored this multi-conditional setting, proposing various strategies to merge guidance signals. Despite their progress, these approaches often struggle with *condition conflicts*, where competing instructions degrade image quality and weaken the model's adherence to the combined user input.

We identify the root of the problem as the *parallel generation* process inherent to diffusion models. At every refinement step, all guidance signals are applied globally and concurrently across the entire image canvas. This forces the model into a "tug-of-war" when conditions provide contradictory information for the same region—for instance, a canny map demanding a sharp outline while a subject condition requires soft fur texture. The model is forced to average these competing signals, leading to a suboptimal compromise.

We argue that a more effective solution requires a paradigm shift from parallel refinement to *serial generation*. This perspective brings autoregressive models into focus, which have re-emerged as a powerful paradigm for visual synthesis (Ramesh et al., 2021; Esser et al., 2021; Yu et al., 2022). These models construct an image sequentially, one token at a time. This inherent sequentiality provides a natural mechanism for resolving conflicting constraints. Rather than forcing a global compromise, an autoregressive model makes a series of localized decisions. As it generates the image patch by patch, it can dynamically prioritize the most salient condition for each specific location—attending to the edge map for a character's outline, for instance, before shifting focus to the subject guidance for its interior texture. This transforms the generation process from a contentious negotiation into a coherent, step-by-step composition.

Building on this insight, we introduce **ContextAR**, a framework designed to unlock the full potential of autoregressive models for multi-conditional image generation. Our contributions are:

- We analyze the core challenges of multi-condition control, clearly articulating why the serial generation process of autoregressive models is inherently better suited to resolving constraint conflicts than the parallel approach of diffusion models.

- We propose ContextAR, an effective and efficient multi-condition controllable generation framework for autoregressive models. By representing all conditions within a unified joint sequence, our method enhances the precision and flexibility of control.

- We provide strong experimental evidence showing that ContextAR surpasses leading diffusion-based methods in complex multi-condition tasks. It demonstrates superior control and faithfulness to user inputs, all while maintaining high image quality.

## 2 RELATED WORK

**Generative Diffusion Models.** Generative models based on continuous-time dynamics have become the de facto standard for high-fidelity image synthesis. This broad category is primarily composed of two major paradigms. The first is the widely adopted denoising diffusion framework (Song et al., 2020; Ho et al., 2020; Rombach et al., 2022), which iteratively denoising a random Gaussian noise map. The second, more recent paradigm is flow matching (Lipman et al., 2022), which learns a direct mapping from noise to data. This approach has also demonstrated remarkable success (Labs, 2024; Esser et al., 2024). Despite their different mathematical foundations, both denoising- and flow-based approaches share a fundamental characteristic: they operate via a parallel refinement process, where the entire image representation is updated at each step. This shared parallel nature is central to our analysis of their limitations in multi-conditional settings.

**Autoregressive Image Generation.** Autoregressive models, long dominant in natural language processing (Vaswani, 2017; Brown et al., 2020), have also demonstrated strong performance in visual synthesis (Esser et al., 2021; Yu et al., 2022; Ramesh et al., 2021; Sun et al., 2024). They treat an image as a 1D sequence of discrete tokens (typically from a VQ-VAE tokenizer (Van Den Oord et al., 2017)) and generate it one token at a time based on previously generated tokens. This serial, token-by-token generation process is fundamentally different from diffusion models. Recent work has significantly improved the quality and efficiency of autoregressive models (Tian et al., 2024; Li et al., 2024b; Yu et al., 2025; Wang et al., 2025b), making them competitive with diffusion models.

**Controllable Image Generation.** Providing users with fine-grained control over generation is a critical area of research. For diffusion models, this has been extensively studied. ControlNet (Zhang et al., 2023) and T2I-Adapter (Mou et al., 2024) enabled precise spatial control using conditions like canny or depth maps. Other methods like IP-Adapter (Ye et al., 2023) focused on subject-driven control. The challenge of combining these has led to multi-conditional frameworks like UniCombine (Wang et al., 2025a), which attempt to merge multiple parallel guidance signals. In contrast, controllable autoregressive generation is less explored. Early methods focused on single conditions, either by simple token concatenation (Mu et al., 2025) or through external feature fusion (Li et al., 2024c; Cao et al., 2024). The problem of multi-conditional control in autoregressive models, and the potential benefits of their serial nature in resolving condition conflicts, has remained largely unaddressed. Our work fills this gap by proposing the first dedicated framework for this task.

## 3 CONDITION CONFLICTS IN MULTI-CONDITION CONTROL

The primary challenge in multi-conditional image generation lies in satisfying multiple, often conflicting, constraints simultaneously. The architectural paradigm of the generative model plays a crucial role in how these constraints are integrated. We categorize current approaches into two paradigms: parallel generation, exemplified by diffusion models, and serial generation, embodied by autoregressive models. Figure 2 schematically illustrates the difference between these two generation methods under a similar model architecture.

**Parallel Generation: Global Conflict.** Diffusion models operate in a parallel manner, where the entire image representation is iteratively refined at each denoising step. When conditions are introduced, their corresponding guidance signals are injected globally and concurrently. As shown in Figure 3(a), under a single condition, there are high attention values between the denoised image and the condition, enabling effective conditional control. However, in multi-condition tasks, this simultaneous application can lead to a "tug-of-war" between conditions. Figures 3(c) illustrate that at each timestep, the model must attend to multiple conditions simultaneously, causing the attention values between the denoised image and the multiple conditions in Figure 3(b) to be dispersed. This results in suboptimal control for each condition, leading to a loss of detail.

**Serial Generation: Local Resolution of Constraints.** In contrast, autoregressive models generate images serially, predicting one token at a time in a predefined order (e.g., raster scan). Under a single condition, the serial method also allows the token to be generated to focus well on the condition image, resulting in effective control. Moreover, this sequential process offers a natural

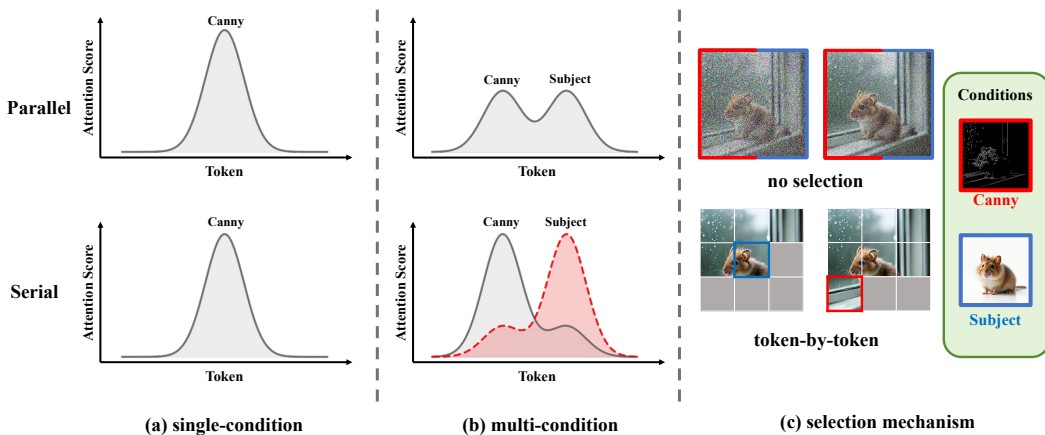

Figure 3: Comparison of performance under single-condition and multi-condition tasks.

mechanism for resolving conflicts in multi-condition tasks. As shown in Figure 3(c), when generating a specific token $q_t$, the model can selectively prioritize the most relevant condition. For example, when generating a part of the image corresponding to the mouse's body, it can prioritize the subject condition to render fine fur details. As it moves to generate the background (the window), it can shift its focus to weigh the canny map more heavily. The information from previously generated tokens ($q_{<t}$) provides context that helps the model make coherent decisions and smoothly transition between regions dominated by different conditions. As shown in Figure 3(b), the model's prioritized attention to different conditions when generating different tokens can avoid the dispersion of attention, leading to more precise control.

## 4 METHOD

Leveraging the theoretical advantages of serial generation, we introduce our framework, **ContextAR**. It enhances a standard autoregressive image generation pipeline by representing diverse conditions within a unified sequence.

### 4.1 AUTOREGRESSIVE GENERATION WITH SPATIALLY OVERLAPPED UNIFIED SEQUENCES

**Autoregressive Image Generation.** Autoregressive models for image generation typically first use a VQ-VAE (Van Den Oord et al., 2017) to encode an image $\mathcal{I}$ into a sequence of discrete tokens $\boldsymbol{q} = (q_1, \ldots, q_N)$. A Transformer-based model is then trained to predict the next token in the sequence, conditioned on the preceding tokens and any other context $c$. The probability is factorized as $p(\boldsymbol{q} \mid c) = \prod_{t=1}^{N} p(q_t \mid q_{<t}, c)$. We use LlamaGen (Sun et al., 2024) as our base model, which prepends text tokens to the image token sequence, allowing for text-to-image synthesis.

**Unified Sequence.** To handle multiple conditions, we extend this unified sequence paradigm. As illustrated in Figure 4(a), we represent text, multiple visual conditions, and the target image as a single, concatenated sequence: $\boldsymbol{S} = [\boldsymbol{c}_1, \ldots, \boldsymbol{c}_m, \boldsymbol{c}_T, \boldsymbol{q}]$, where each $\boldsymbol{c}_k$ is the token sequence for a visual condition and $\boldsymbol{c}_T$ is for the text prompt. Each condition type has its own embedding layer, initialized from the image embedding layer weights, to capture condition-specific features while sharing a common semantic space.

**Hybrid Positional Embeddings.** Precise spatial control requires careful handling of positional information. Naïvely applying the same 2D Rotary Position Embedding (RoPE) (Su et al., 2024) to all visual tokens (conditions and image) ensures spatial alignment, which is especially effective for spatially-aligned conditions. However, this makes it difficult for the model to distinguish between different condition types at the same location. To resolve this, we propose a hybrid approach. As shown in Figure 4(b), we first apply the same 2D RoPE to all visual tokens. Then, we add a unique,

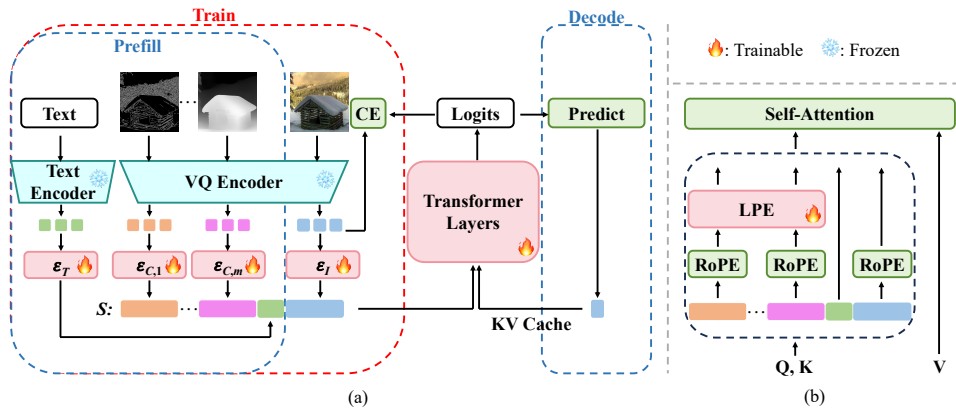

Figure 4: Overview of our proposed ContextAR. (a) The overall process of training and inference. Visual conditions, text, and images are incorporated into unified sequence processing. (b) Positional embedding before attention operations.

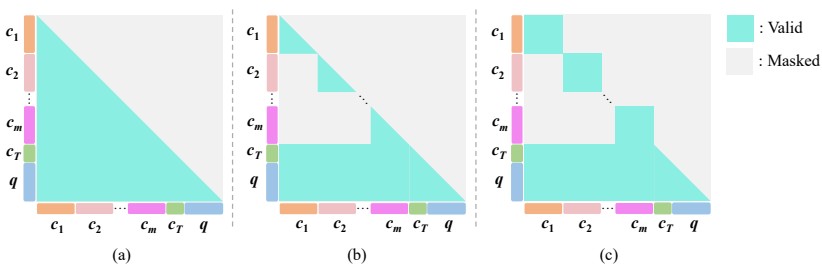

Figure 5: Visualization of attention computation. (a) Normal attention with causal mask, (b) Cross-Condition Perception Restriction, (c) Intra-Condition Bidirectional Perception.

condition-specific Learnable Positional Embedding (LPE) $P_k$ to the query and key of each condition token sequence $c_k$. This provides a distinct positional signature for each condition type, allowing the model to differentiate them while maintaining spatial correspondence with the target image.

## 4.2 CONDITIONAL CONTEXT-AWARE ATTENTION

**Cross-Condition Perception Restriction (CCPR).** To ensure that conditions can be flexibly combined at inference time, we prevent direct attention between different condition types. As shown in Figure 5(b), we mask the attention scores between queries of one condition type and keys of another. A significant benefit is the reduction in computational complexity of the train and prefill stage from quadratic to linear with respect to the number of conditions.

**Intra-Condition Bidirectional Perception (ICBP).** While image tokens must be generated autoregressively, the condition tokens are fully available at the start of the process. Standard causal attention is therefore unnecessary for processing conditional inputs. Inspired by vision transformers (Xie et al., 2024), we allow bidirectional attention **within** each condition sequence (Figure 5(c)). This enables each condition token to see all other tokens of the same type, improving the model's spatial understanding of each condition independently before using it to guide image generation.

## 4.3 TRAINING AND INFERENCE

**Training.** We train the model using a standard next-token prediction loss on the image tokens $q$. To improve robustness and enable classifier-free guidance, we randomly drop the text condition (with probability 0.1) and each visual condition (with probability 0.25) during training by masking them out from the attention computation.

**Inference.** Generation is a two-stage process. In the **prefill** stage, all condition sequences are processed in parallel to compute their KV caches. In the **decode** stage, image tokens are generated one by one autoregressively, attending to the cached conditions. Thanks to CCPR and our training strategy, any subset of the trained conditions can be used during inference without fine-tuning. Moreover, we can use classifier-free guidance to enhance controllability.

**Implementation for Acceleration.** To realize the computational benefits of CCPR, our practical implementation for both training and inference differs from a literal attention mask. Instead, we partition the unified sequence into blocks according to condition type and compute self-attention only **within** each block. This approach directly avoids the expensive cross-condition computations, thereby achieving the intended acceleration.

## 5 EXPERIMENTS

We conduct a series of experiments to validate our central hypothesis: that a serial, autoregressive approach offers superior controllability and flexibility in multi-conditional image generation compared to parallel, diffusion-based methods. We evaluate our proposed ContextAR against state-of-the-art multi-condition diffusion models and other autoregressive baselines.

### 5.1 SETUP

**Datasets and Baselines.** We conduct experiments on MultiGen-20M (Qin et al., 2023) for compositional space-aligned conditions (canny, depth, hed, pose) and compare with several state-of-the-art diffusion-based multi-conditional frameworks (Zhao et al., 2023; Qin et al., 2023; Hu et al., 2023; Pan et al., 2025). We also compare our method with the autoregressive-based conditional control approach ControlAR (Li et al., 2024c), despite its single-condition limitation. To further validate our method, following UniCombine (Wang et al., 2025a), we perform experiments on SubjectSpatial200K combining subject-driven and spatially-aligned conditions (subject-canny, subject-depth).

**Metrics.** We compute FID (Heusel et al., 2017) and MUSIQ (Ke et al., 2021) metrics to evaluate the quality of the generated images. Additionally, we employ SSIM (Wang et al., 2004) to measure the similarity between the generated images and real images, which serves to validate the controllability. For specific condition types, we further utilize dedicated metrics to assess the conditional control capability. In particular, we extract features from the generated images and compute the F1 score for canny, MSE for depth and SSIM for hed, against their respective condition images. For subject conditions, we calculate the CLIP-I (Radford et al., 2021) score between the generated images and the reference subject images to evaluate their alignment.

**Implementation Details.** We use LlamaGen-XL (Sun et al., 2024) as our base autoregressive model. All training is done on NVIDIA A100 40GB GPUs. For MultiGen-20M, we use 3 GPUs with a batch size of 1 each and set gradient accumulation to 16 steps, yielding an effective batch size of 48. For SubjectSpatial200K, we use 2 GPUs with a batch size of 2 each and set gradient accumulation to 8 steps, giving a total batch size of 32. We use the AdamW optimizer with a learning rate of 5e-5. Training consists of 30,000 iterations on MultiGen-20M and 25,000 iterations on SubjectSpatial200K. All images are 512×512 pixels. The CFG scale is set to 3.0.

### 5.2 MAIN RESULTS

**Multi-Spatial Conditions** Table 1 presents the primary results on the MultiGen-20M dataset, comparing our method against several leading diffusion-based frameworks. The results strongly support our claims. ContextAR achieves the highest SSIM score by a significant margin (a 21.55% improvement over the next best), directly validating its superior ability to adhere to multiple spatial conditions simultaneously. This suggests that the serial generation process is indeed more effective at avoiding the condition conflicts that can plague parallel models. Furthermore, our method attains the best FID score, indicating that this enhanced controllability comes without a trade-off in image quality. Notably, when compared to PixelPonder, which is based on the advanced diffusion model FLUX.1-dev (Labs, 2024), our method achieves comparable generation quality despite the

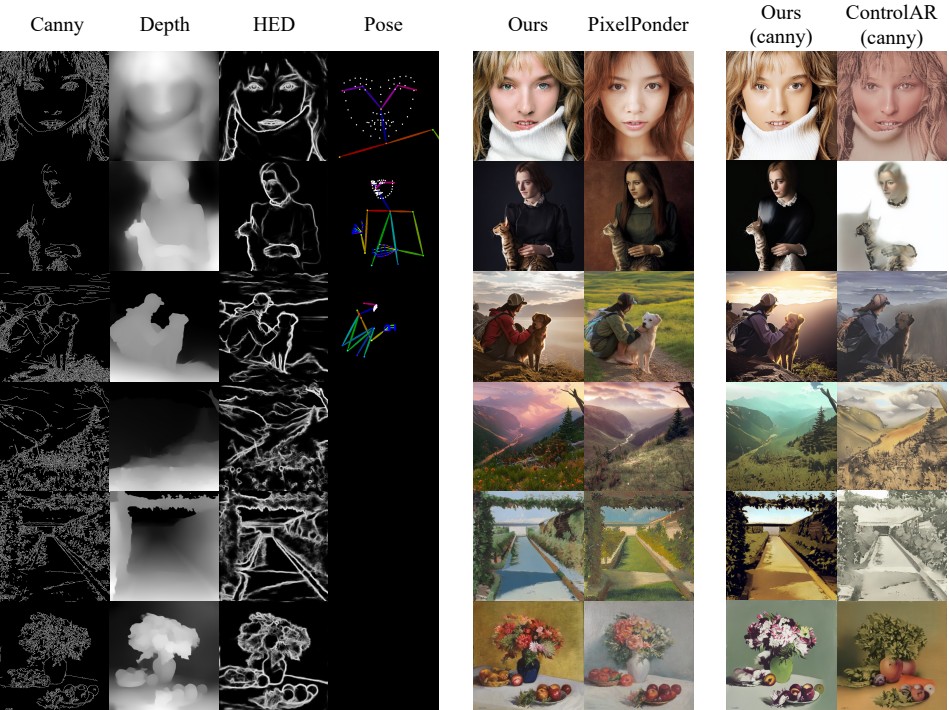

Figure 6: Visualization comparison on MultiGen-20M.

underlying autoregressive model having significantly fewer parameters than FLUX.1-dev (775M vs 12B). Figure 1(a) and Figure 6 visualize the generated results under multiple spatial conditions. Images generated by our method faithfully adhere to various conditional information simultaneously, resulting in high similarity to the referenced original images.

Table 1: Comparison of generation quality and controllability on the MultiGen-20M dataset. The best results are bolded, and the second-best results are underlined. "all" indicates that all conditions (canny, depth, hed, pose) are used.

| Methods | Base Model | Conditions | Metrics | | |
|---|---|---|---|---|---|
| | | | FID ↓ | SSIM ↑ | MUSIQ ↑ |
| Uni-ControlNet | SD1.5(860M) | all | 32.58 | 29.37 | 65.85 |
| UniControl | SD1.5(860M) | all | 25.15 | 35.58 | **72.05** |
| Cocktail | SD2.1(860M) | pose+hed | 24.67 | 24.14 | 67.13 |
| PixelPonder | FLUX.1-dev(12B) | all | 11.85 | 43.99 | 69.54 |
| Ours | LlamaGen-XL(775M) | all | **10.42** | **53.47** | 70.35 |

**Comparision with ControlAR.** Our method differs from another autoregressive-based control method ControlAR (Li et al., 2024c), which is a single-condition method, meaning that only one condition type can be used at a time. In Table 2, we directly compare our model trained in a multi-conditional setting with ControlAR by using only a single condition type as input. As shown, even without specific training for individual conditions, our method still outperforms ControlAR. Furthermore, we demonstrate that combining multiple condition types maintains high control capabilities across condition-specific metrics while achieving consistent FID reductions as more conditions are added. This confirms the flexibility of our approach and the effectiveness of our joint conditional control mechanism. Figure 1(b)(c)(d)(e) and Figure 6 shows the generation results of our method under various condition combinations.

Table 2: Comparison of generation quality and controllability on the MultiGen-20M dataset. Our method and ControlAR are both based on the same LlamaGen-XL model. Note that our model is jointly trained on four conditions (canny, depth, hed, pose) on MultiGen-20M, and can directly select a subset of conditions during inference without fine-tuning for specific condition combinations.

| Methods | Conditions | Metrics | | |
|---|---|---|---|---|
| | | FID ↓ | F1 ↑ | SSIM(hed) ↑ |
| ControlAR | canny | 30.44 | 0.31 | - |
| Ours | canny | **18.53** | **0.34** | - |
| ControlAR | hed | 12.62 | - | 83.48 |
| Ours | hed | **11.86** | - | **83.52** |
| Ours | canny+hed | 11.09 | 0.33 | 83.45 |
| Ours | all | 10.42 | 0.33 | 83.53 |

Subject    Canny    Ours    UniCombine       Subject    Depth    Ours    UniCombine

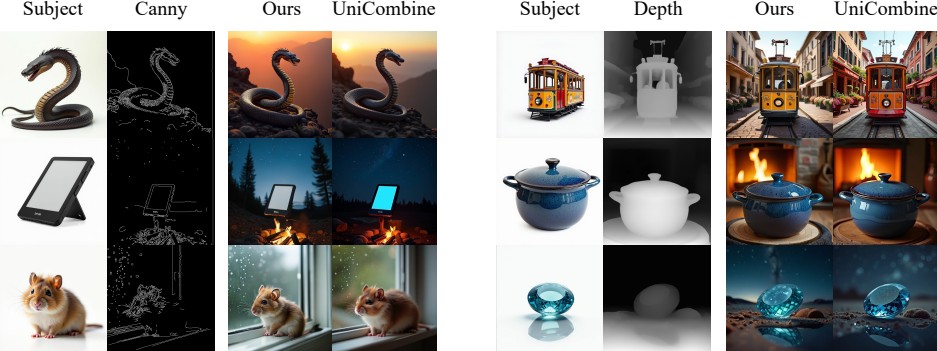

Figure 7: Visualization comparison on SubjectSpatial200K.

**Subject-Spatial Conditions.** Subject-driven conditions differ from spatially-aligned conditions in that they do not require precise point-to-point spatial control, but rather semantic alignment. In Table 3, we compare our method with UniCombine under the combined subject-canny and subject-depth conditions. The F1, MSE, and CLIP-I scores demonstrate that our method can simultaneously maintain high semantic consistency and spatial alignment. Figure 1(f)(g) and Figure 7 showcase our method's generation results for this task. Our approach is effective for both spatial-type conditions and subject-type conditions, demonstrating high generality.

Table 3: Comparison of generation quality and controllability on the SubjectSpatial200K dataset.

| Methods | Base Model | Conditions | Metrics | | | |
|---|---|---|---|---|---|---|
| | | | FID ↓ | F1 ↑ | MSE ↓ | CLIP-I ↑ |
| UniCombine | FLUX.1-schnell | subject+canny | **6.01** | 0.24 | - | 78.92 |
| Ours | LlamaGen-XL | subject+canny | 6.62 | **0.30** | - | **79.27** |
| UniCombine | FLUX.1-schnell | subject+depth | 6.66 | - | 196.65 | 79.01 |
| Ours | LlamaGen-XL | subject+depth | **6.55** | - | **165.56** | **79.22** |

## 5.3 ABLATION STUDY

**CFG Scale.** We evaluated the impact of different CFG scale values on controllability and generation quality. As shown in Table 4, the SSIM increases with larger CFG scales, indicating improved conditional control. This validates the effectiveness of our condition drop strategy during training. However, excessively large CFG values lead to degraded generation quality. Therefore, we select CFG = 3.0 to balance controllability and image fidelity.

Table 4: Comparison of differnet CFG scale on the MultiGen-20M dataset. Here all conditions (canny, depth, hed, pose) are used.

| CFG | Metrics | | |
|-----|---------|--------|----------|
| | FID ↓ | SSIM ↑ | MUSIQ ↑ |
| 1.0 | 9.29 | 50.02 | 69.61 |
| 2.0 | 9.79 | 52.94 | 70.28 |
| 3.0 | 10.42 | 53.47 | 70.35 |
| 4.0 | 10.85 | 53.65 | 70.32 |
| 7.5 | 11.46 | 53.66 | 70.33 |

Table 5: Ablation of positional embedding on the SubjectSpatial200K dataset. Here the conditions used are subject and depth.

| Methods | Metrics | | |
|---------|---------|--------|----------|
| | FID ↓ | MSE ↓ | CLIP-I ↑ |
| RoPE | 6.67 | 167.09 | 79.11 |
| RoPE+LPE | 6.55 | 165.56 | 79.22 |

**Positional Embedding.** To validate the effectiveness of our RoPE+LPE approach for condition images, we compared this strategy with simply applying the same RoPE to both the condition sequence $c_k$ and the image sequence $q$. The results are shown in Table 5. As observed, the RoPE+LPE strategy achieves more precise control.

**Attention Design.** We evaluated three attention mechanisms (standard causal attention, CCPR, and CCPR+ICBP) in terms of per-step computation speed during the training phase. As shown in Figure 8, as the number of conditions increases, the computational cost with the CCPR mechanism is significantly lower than that of standard attention, demonstrating the effectiveness of our approach. Moreover, the bidirectional attention introduced in ICBP adds almost no additional computational overhead. Furthermore, we examined the impact of ICBP on conditional control capability. The results in Figure 9 show that incorporating the ICBP mechanism improves controllability, confirming that bidirectional attention is beneficial for understanding visual modality conditional information.

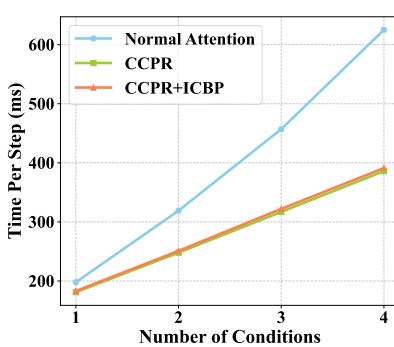

Figure 8: Comparison of different attention mechanisms on the training time cost. Here the batch size is 1.

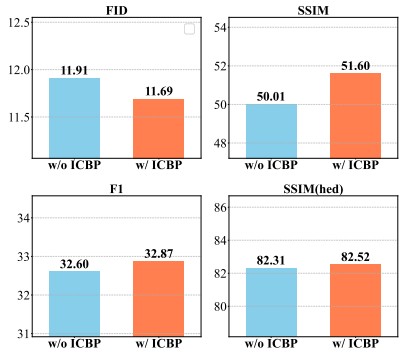

Figure 9: Ablation of ICBP on the MultiGen-20M dataset. Here all conditions are used. Both methods are trained for 10k iterations.

## 6 CONCLUSION

In this paper, we arguing that the parallel processing of diffusion models leads to inherent condition conflicts in multi-conditional image generation. We proposed a paradigm shift towards serial, autoregressive generation, which can resolve conflicting constraints locally. Our proposed framework, **ContextAR**, operationalizes this idea by integrating multiple conditions into a unified sequence processed by an efficient, context-aware attention mechanism. Experimental results confirm our hypothesis, demonstrating that ContextAR outperforms state-of-the-art diffusion-based methods in controllability and faithfulness to combined conditions, without compromising image quality. This work highlights the promise of autoregressive models as a powerful and flexible foundation for future research in complex, multi-conditional generative tasks.

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

## A  DATASET DETAILS

For experiments on MultiGen-20M, we randomly sampled approximately 700,000 images from the full dataset to construct our training set. To ensure a fair comparison with PixelPonder Pan et al. (2025), we followed the same evaluation protocol by merging the original validation set (5,000 images) and test set (500 images) of MultiGen-20M, resulting in a test set comprising 5,500 samples. The following standard algorithms were applied to each image to generate various control conditions: Canny edge maps using Canny (1986), depth maps using Ranftl et al. (2020), HED edge maps using Xie & Tu (2015), and pose maps using Yang et al. (2023).

For experiments on SubjectSpatial200K, we adopted the official train/test split as recommended by UniCombine Wang et al. (2025a), which consists of 139,403 training samples and 5,827 test samples.

## B  COMPUTATIONAL RESOURCES

All experiments were conducted on NVIDIA A100 40GB GPUs. For training on the MultiGen-20M dataset, we used 3 GPUs in parallel for 30,000 iterations, with the total wall-clock time being 54 hours, resulting in 162 GPU-hours. For the SubjectSpatial200K dataset, we carried out two separate runs (subject+depth and subject+canny), each using 2 GPUs for 25,000 iterations and approximately 24 hours per run, totaling 48 GPU-hours per run and 96 GPU-hours overall. All inference experiments were performed on a single A100 GPU. Specifically, inference on the validation set for 4-condition combinations required 5 hours (5 GPU-hours), while inference for 2-condition combinations required 3 hours (3 GPU-hours).

## C  MORE EXPERIMENTS

### C.1  EVALUATION ON SINGLE-CONDITION TASKS

We benchmarked our model and several baselines across multiple single-condition tasks. Notably, EditAR (Mu et al., 2025) and ControlAR (Li et al., 2024c) are autoregressive methods that share

the same base model architecture (LlamaGen-XL) as our approach. As illustrated in Table 6, our approach also achieves strong controllability across diverse single-condition settings.

Both EditAR and our method use pre-filled condition sequences for controllable autoregressive generation. However, our approach demonstrates superior controllability with the same backbone and comparable training steps. The key differences are: (1) **Positional Encoding**: EditAR uses non-overlapping positional offsets for conditions and target images, while we employ spatially overlapping RoPE. (2) **Embedding Layers**: EditAR shares a single embedding layer across all token types, while we use dedicated embeddings for each condition type and target images.

Table 6: Evaluation on single-condition tasks. We use COCO-Stuff dataset (Caesar et al., 2018) for seg and MultiGen-20M dataset for canny, depth and HED.

| Method | Canny | Depth | HED | Segmentation |
|---|---|---|---|---|
| | (F1 ↑) | (RMSE ↓) | (SSIM(hed) ↑) | (mIoU ↑) |
| Uni-ControlNet (Zhao et al., 2023) | 0.27 | 40.65 | 69.10 | 31.60 |
| ControlNet++ (Li et al., 2024a) | **0.37** | 28.32 | 80.97 | 34.56 |
| EditAR (Mu et al., 2025) | 0.25 | 34.93 | - | 22.62 |
| ControlAR (Li et al., 2024c) | 0.31 | 29.01 | 83.48 | **37.49** |
| Ours | 0.34 | **26.73** | **83.52** | 36.36 |

## C.2 Comparison with SOTA Diffusion-based Control Models

Furthermore, we conducted comparisons with several state-of-the-art diffusion-based control models, including FLUX.1-dev-ControlNet-Union-Pro-2.0 (Shakker-Labs, 2025), FLUX.1-canny (Labs, 2024), UNO (Wu et al., 2025), and Bagel (Deng et al., 2025).

Table 7: Comparison with SOTA diffusion models on MultiGen-20M under the canny condition.

| Method | Base Model | FID ↓ | F1 ↑ |
|---|---|---|---|
| FLUX.1-dev-ControlNet-Union-Pro-2.0 | FLUX.1-dev(12B) | **16.83** | 0.27 |
| FLUX.1 canny | FLUX.1-dev(12B) | 21.16 | 0.21 |
| Ours | LlamaGen-XL(775M) | 18.53 | **0.34** |

Table 8: Comparison with SOTA diffusion models on SubjectSpatial200K under the subject condition.

| Method | Base Model | FID ↓ | CLIP-I ↑ |
|---|---|---|---|
| UNO | FLUX.1-dev(12B) | 10.59 | **81.64** |
| Bagel | Bagel(7B) | 10.78 | 81.23 |
| Ours | LlamaGen-XL(775M) | **8.96** | 78.55 |

Notably, our base model (LlamaGen-XL, 775M parameters) is substantially smaller than FLUX.1 (12B) and Bagel (7B). This demonstrates that the controllability and performance of our method do not merely stem from a larger backbone, but from the proposed approach itself.

## C.3 More Ablation Studies

We designed and trained two new variants to further analyze our approach. In **Variant A**, we removed the Cross-Condition Perception Restriction (CCPR), allowing conditions to attend to each other. In **Variant B**, we introduced a sparser training regimen by randomly dropping conditions. Specifically, we used a uniform distribution over the number of conditions from zero to three (i.e., 25% probability for each case of using 0, 1, 2, or 3 conditions).

We trained the original method, Variant A, and Variant B for the same number of steps and evaluated them on MultiGen20M. The 4-condition generation results in Table 9 show that Variant A provides

minimal performance gains despite adding inter-condition attention, validating our design choice to remove this attention for computational efficiency. Variant B shows reduced SSIM in the 4-condition scenario due to its sparse-condition training.

Table 9: Ablation study on MultiGen20M with 4 conditions.

| Variant | FID $\downarrow$ | SSIM $\uparrow$ | MUSIQ $\uparrow$ |
|---|---|---|---|
| Original | 11.96 | 49.52 | 70.18 |
| A (w/o CCPR) | 12.02 | 49.64 | 70.25 |
| B (Sparse Training) | 11.84 | 48.47 | 69.79 |

Further evaluation of Variant B under single canny conditions (Table 10) shows it underperforms in F1 and SSIM metrics, with only FID showing slight improvement. This confirms our original training method effectively handles both multi-condition and sparse-condition scenarios.

Table 10: Evaluation of Variant B on single canny condition.

| Variant | FID $\downarrow$ | F1 $\uparrow$ | SSIM $\uparrow$ | MUSIQ $\uparrow$ |
|---|---|---|---|---|
| Original | 23.48 | 32.88 | 38.60 | 68.19 |
| B (Sparse Training) | 20.15 | 32.51 | 37.25 | 68.47 |

## D  STATISTICAL SIGNIFICANCE

To assess the robustness of our experimental results, we evaluate metrics across 5 different random seeds, reporting the 95% confidence intervals in Table 11 and Table 12. The consistently low variances observed indicate that our method achieves stable and reliable performance across multiple runs. This demonstrates that the improvements reported are statistically significant and not due to random chance.

Table 11: The 95% confidence intervals of FID, SSIM, and MUSIQ metrics on the MultiGen-20M dataset, tested on 5 different random seeds. All conditions are used.

| FID | SSIM | MUSIQ |
|---|---|---|
| $10.45 \pm 0.046$ | $53.51 \pm 0.052$ | $70.33 \pm 0.014$ |

## E  LLM USAGE

To enhance the quality of this manuscript, we utilized a Large Language Model (LLM) for assistance with writing and editing. The primary role of the LLM was to improve the clarity, readability, and overall linguistic style of the text. This involved tasks such as refining sentence structures, correcting grammatical errors, and ensuring a coherent narrative throughout the paper.

We must emphasize that the scope of the LLM's contribution was strictly limited to language enhancement. The conceptualization of the research, including the core ideas, experimental procedures, and data interpretation, was carried out exclusively by the authors. The LLM had no part in the scientific aspects of this work.

The authors retain full accountability for all content within this paper, including any text that was revised or improved with the assistance of the LLM. We confirm that its use complies with all ethical standards regarding academic integrity and originality.

## F  LIMITATIONS

While **ContextAR** demonstrates strong flexibility and controllability for multi-conditional image generation, several aspects remain open for future exploration.

Table 12: The 95% confidence intervals of FID, MSE, and CLIP-I metrics on the SubjectSpatial200K dataset(subject+depth condition), tested on 5 different random seeds.

| FID | MSE | CLIP-I |
|---|---|---|
| $6.57 \pm 0.03$ | $164.56 \pm 3.50$ | $79.19 \pm 0.03$ |

First, while our framework is built upon a representative autoregressive backbone, recent progress has led to even more advanced autoregressive architectures. Evaluating and extending ContextAR on top of stronger or larger-scale autoregressive models remains a promising avenue for future work.

Second, while ContextAR has been validated on diverse spatially-aligned and subject-driven conditions, its performance on other complex or highly abstract modalities—such as 3D structure or video sequences—remains to be systematically studied. Incorporating more heterogeneous modalities would be a promising direction for broadening the framework's applicability.

Finally, despite significant improvements in computational efficiency introduced by our attention design, the overall inference speed of large autoregressive models can still be a limiting factor in latency-sensitive applications. Further research on model acceleration, compression, or more efficient architectures could help to address this challenge.

We believe addressing these directions will further enhance the versatility and impact of autoregressive multi-conditional generation models.

## G  SOCIAL IMPACT

Our work aims to advance the field of controllable image generation by improving the flexibility and generality of autoregressive models for multi-conditional scenarios. This technological progress has the potential to benefit a wide range of creative, educational, and scientific applications, such as computer-aided design, digital art creation, virtual reality, and assistive technologies for content generation.

Nevertheless, as with many generative models, there is a possibility that the proposed framework could be misused for generating synthetic images that are misleading or violate privacy, intellectual property, or ethical guidelines. To mitigate such risks, we recommend responsible deployment practices, including proper content filters, usage restrictions, and clear disclosure of synthetic content. We encourage the research community and downstream users to carefully consider ethical implications and abide by relevant laws and standards when applying such models.

Overall, we believe that the positive impacts of flexible, controllable image generation frameworks can be maximized through responsible research and deployment, while proactively addressing potential negative consequences.

## H  MORE VISUAL RESULTS

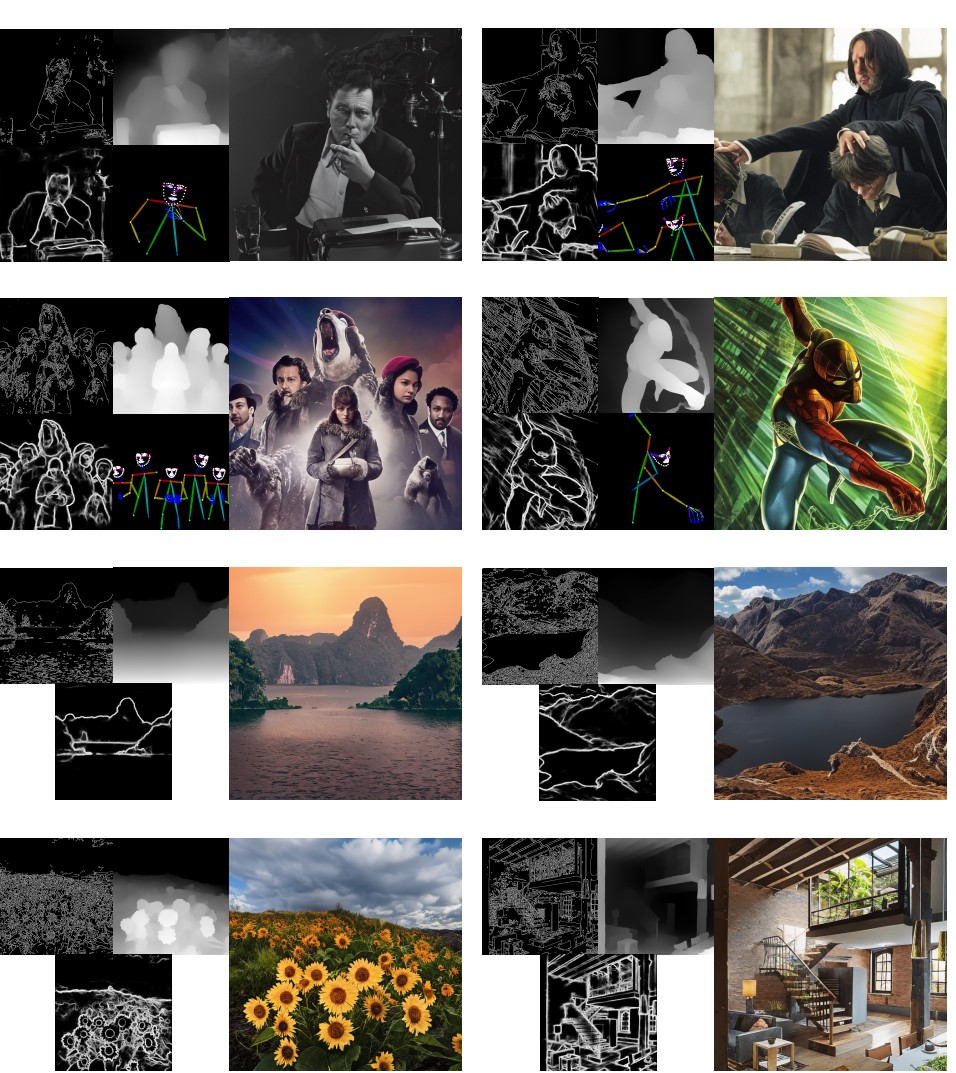

Figure 10: More visual results on the MultiGen-20M dataset.

