# OpenReview forum: "Context-Aware Autoregressive Models for Multi-Conditional Image Generation"
_ICLR.cc/2026/Conference — ICLR 2026 Conference Withdrawn Submission_

### Official Review · Reviewer_3rqa · 2025-10-15

**Soundness:** 1
**Presentation:** 1
**Contribution:** 1
**Rating:** 2
**Confidence:** 4

**Summary:**

The authors propose ContextAR, an autoregressive image generation model that can take in multiple (dense) conditional images (e.g. pose, canny edge maps, etc.).

**Strengths:**

ControlAR outperforms previous baselines on MultiGen-20M.

**Weaknesses:**

1. My largest reservation is that the method is poorly motivated. All of section 3 is vague. At best, this is a hypothesis with no backup theory or empirical validation.

1-1. The main claim for switching from parallel generation to sequential generation is that the conditions provide contradictory information for the same region. This also holds true for autoregressive models as the conditions are given in the same way. Generating an image patch-by-patch does not prevent this from happening.

1-2. To backup the claim, the authors point to figure 3, which is not an experimental figure. This is conceptual, which does not provide evidence.

2. Limited novelty: The method uses a standard autoregressive architecture formulation with multiple conditions.

3. In table 1, the authors show that the proposed method outperforms previous methods, with smaller number of parameters. While this is appealing, a better comparison would be to train a (parallel) diffusion model with the same architecture (with modifications in the attention), with matched compute to truly see the advantage. Even better would be to show the scaling property of both methods - which scales better?

**Questions:**

Please see weaknesses.

---

### Official Review · Reviewer_EQNZ · 2025-10-31

**Soundness:** 1
**Presentation:** 1
**Contribution:** 1
**Rating:** 2
**Confidence:** 4

**Summary:**

This paper addresses the challenge of multi-conditional image generation, where multiple input constraints (e.g., canny edges, depth, pose, or subject identity) must be harmonized to produce a single coherent image. Existing diffusion-based approaches often suffer from condition conflicts, where competing guidance signals degrade image fidelity or cause inconsistent adherence to conditions.

The authors attribute this issue to the parallel generation process in diffusion models, which applies all conditions globally and simultaneously at each denoising step. To overcome this, they propose shifting to serial (autoregressive) generation, arguing that token-by-token synthesis can resolve local conflicts more naturally.

They introduce ContextAR, an autoregressive framework that concatenates multiple condition tokens (encoded via VQ-VAE) into a unified sequence and processes them through a transformer equipped with a Conditional Context-Aware Attention mechanism. This includes Cross-Condition Perception Restriction (CCPR) to block inter-condition attention and Intra-Condition Bidirectional Perception (ICBP) to improve spatial coherence within each condition. Experiments on MultiGen-20M and SubjectSpatial200K demonstrate improved controllability and competitive FID compared to diffusion-based baselines like UniControl, Cocktail, and PixelPonder.

**Strengths:**

**Empirical performance**: The method achieves strong controllability (highest SSIM) and comparable image quality (FID) to state-of-the-art diffusion systems despite using a smaller AR backbone.

**Unified formulation**: The design supports flexible combinations of conditions during inference without retraining, highlighting modularity in condition composition.

**Weaknesses:**

**Limited model novelty.**

The architectural contribution appears incremental. The proposed ContextAR primarily concatenates multiple condition tokens (encoded via VQ-VAE) into a unified sequence and slightly modifies the attention pattern to restrict cross-condition interaction. Such conditioning strategies have already been explored in ControlAR (Li et al., 2024c, ICLR’25) and EditAR (Mu et al., 2025, arXiv’25). As a result, the novelty lies mostly in implementation refinements rather than in introducing a fundamentally new modeling principle or generative paradigm.

**Limited flexibility.**

The method requires retraining a full autoregressive transformer (LlamaGen-XL) to support multiple conditions, whereas diffusion-based control frameworks (e.g., ControlNet, T2I-Adapter) can often be applied as lightweight plug-in modules. This reduces its practicality and contradicts the paper’s claim of offering a flexible and general multi-condition generation framework.

**No quantitative evaluation of condition conflict resolution.**

The central claim, that serial autoregressive generation inherently resolves condition conflicts, is not empirically substantiated. The paper provides only qualitative visualizations, without any quantitative metric or analysis (e.g., per-condition attention overlap or a condition consistency score) to verify conflict mitigation. This weakens the validity of its main motivation.

**Questions:**

Q1. How exactly does the proposed serial generation resolve conflicts rather than merely distribute them temporally? Wouldn’t attention at each step still aggregate multiple conflicting condition signals?

Q2. How scalable is the unified-sequence formulation? As the number or resolution of conditions increases, does the token sequence length and attention cost grow prohibitively?

Q3. Could the authors compare with other AR-based conditional methods (e.g., EditAR, ControlAR) under identical datasets and conditions to isolate the gain from the proposed attention design?

Q4. Can the model support dynamically weighted or adaptive condition fusion (e.g., prioritizing one condition over another) without retraining?

---

### Official Review · Reviewer_jPUv · 2025-11-03

**Soundness:** 1
**Presentation:** 2
**Contribution:** 2
**Rating:** 2
**Confidence:** 5

**Summary:**

This study introduces ContextAR for autoregressive image generation conditioned on multiple conditions. The authors claim that diffusion models inherently fail to leverage multiple conditions due to the issue of tug-of-war and autoregressive models can resolve the issue. Fine-tuning an image generation model conditioned on multiple conditions shows remarkable performance compared with previous frameworks after adding learnable positional embedding and the attention mask for intra-condition bidirectional perception.

**Strengths:**

S1. The proposed methods, learnable positional embedding and various attention masks for effectively leveraging multiple conditions, make sense.

S2. Empirical results show promising and competitive performance.

**Weaknesses:**

Although I think the experiments present promising results, I have serious concerns regarding the rationale behind the key claims and experiments.

W1. The key motivation lacks theoretical or empirical support. The paper claims that diffusion models inherently suffer from a “tug-of-war” between conditions, whereas autoregressive models can resolve this issue. However, there is neither theoretical nor empirical analysis to support this claim, since Section 3 is merely based on an assumption. In fact, this assumption may be incorrect — although diffusion models denoise all latent vectors jointly, each latent vector can attend to multiple condition tokens through multi-head attention. Because of positional embeddings, each noised latent vector can sufficiently attend to the corresponding local information of each conditioning image, so the image condition does not necessarily serve as a global guidance for the diffusion model. Moreover, diffusion models can leverage diffusion time steps to condition on multiple images: high-level noises attend to semantic-level conditions (coarse information), while low-level noises attend to pixel-level conditions (fine-grained information), which is different from autoregressive models. Considering these different behaviors, the authors should conduct a thorough analysis to justify the assumption in Section 3, given that it forms the key motivation for adopting autoregressive models as the main contribution of this paper.

W2. There is no specific experiment that fairly compares diffusion and autoregressive models. Therefore, it remains difficult to substantiate the authors’ main claim — that autoregressive models perform better than diffusion models under multiple conditioning images.

**Questions:**

Please refer to the weakness above. My major concerns are based on W1 and W2. I suggest resolving my major concerns above can improve the overall logic and make this paper more strong.

---

### Official Review · Reviewer_hoUq · 2025-11-04

**Soundness:** 3
**Presentation:** 3
**Contribution:** 3
**Rating:** 6
**Confidence:** 3

**Summary:**

The authors observe that the parallel generation process in mainstream diffusion models leads to condition conflicts when multiple controls are applied. They argue for a paradigm shift to serial generation, which can resolve these conflicts locally.
Based on this insight, the paper proposes ContextAR. The "Context" here refers to two aspects: the "Conditional Context" (i.e., the user-provided controls like Canny or Subject) and the "Sequential Context" (the sequence of already-generated tokens, $q_{<t}$). The core "Context-Aware" capability of the model lies in its use of the sequential context ($q_{<t}$) to dynamically and locally decide which conditional context (e.g., Canny vs. Subject) to prioritize at each generation step.

**Strengths:**

* This paper is well-motivated. In practical applications, users often wish to impose multiple constraints simultaneously (e.g., specifying pose, depth, and subject appearance). The paper clearly identifies a key limitation of parallel generation in diffusion models: when guidance signals from different conditions (like Canny edges and subject textures) are applied globally and concurrently, they can conflict, forcing the model to produce a sub-optimal, "compromised" result. The core idea of using autoregressive generation to shift this "global conflict" into a series of "local decisions" is intuitive and interesting. The model can, for instance, prioritize the Canny condition when generating an outline and then shift its focus to the Subject condition for the internal texture.
* A novel aspect of the methodology is the Conditional Context-aware Attention mechanism. Specifically, the Cross-Condition Perception Restriction (CCPR) prevents attention between different condition types (like Canny and Depth).

**Weaknesses:**

**Extensibility to New Conditions:** A primary weakness concerns the extensibility to new, unseen conditions. The paper's method relies on jointly training all conditions (Canny, Depth, HED, Pose, etc.) as part of a unified sequence. This contrasts with the models like ControlNet, which allow for "plug-and-play" training and addition of new control types onto a frozen base model. The paper does not explicitly discuss how a new condition (e.g., "Scribble" or "Segmentation") could be efficiently added to an already-trained ContextAR model, which would be a significant limitation if retraining from scratch is required.

**Questions:**

* A key design choice in the framework is the Cross-Condition Perception Restriction (CCPR), which explicitly forbids attention between different condition types. The authors justify this as a way to improve flexibility and computational efficiency. However, might this assumption be too rigid? In cases where different conditions have a high degree of synergy (for example, a "Pose" map and a "Canny" map that both describe the same person), could allowing some form of cross-condition attention lead to a more coherent result?
* The ablation study (Variant A, w/o CCPR) shows only a "minimal performance gain" when this restriction is removed.  Does this finding indicate that the model simply failed to learn how to effectively utilize this cross-condition information during training (perhaps as a side-effect of the random condition dropping strategy)?

---

### Note · Authors · 2025-11-12

I have read and agree with the venue's withdrawal policy on behalf of myself and my co-authors.